# The Neurotrophin Receptor TrkC as a Novel Molecular Target of the Antineuroblastoma Action of Valproic Acid

**DOI:** 10.3390/ijms22157790

**Published:** 2021-07-21

**Authors:** Simona Dedoni, Luisa Marras, Maria C. Olianas, Angela Ingianni, Pierluigi Onali

**Affiliations:** 1Laboratory of Cellular and Molecular Pharmacology, Section of Neurosciences, University of Cagliari, 09042 Monserrato, Italy; dedoni@unica.it (S.D.); mariolina.olianas@gmail.com (M.C.O.); 2Section of Microbiology, Department of Biomedical Sciences, University of Cagliari, 09042 Monserrato, Italy; luisamarras@yahoo.it (L.M.); ingianni@unica.it (A.I.)

**Keywords:** valproic acid, HDAC inhibitors, TrkC, MAP kinases, RUNX3, Egr1, p75NTR, neurotrophin-3, apoptosis, human neuroblastoma cells

## Abstract

Neurotrophins and their receptors are relevant factors in controlling neuroblastoma growth and progression. The histone deacetylase (HDAC) inhibitor valproic acid (VPA) has been shown to downregulate TrkB and upregulate the p75NTR/sortilin receptor complex. In the present study, we investigated the VPA effect on the expression of the neurotrophin-3 (NT-3) receptor TrkC, a favorable prognostic marker of neuroblastoma. We found that VPA induced the expression of both full-length and truncated (TrkC-T1) isoforms of TrkC in human neuroblastoma cell lines without (SH-SY5Y) and with (Kelly, BE(2)-C and IMR 32) *MYCN* amplification. VPA enhanced cell surface expression of the receptor and increased Akt and ERK1/2 activation by NT-3. The HDAC inhibitors entinostat, romidepsin and vorinostat also increased TrkC in SH-SY5Y, Kelly and BE(2)-C but not IMR 32 cells. TrkC upregulation by VPA involved induction of RUNX3, stimulation of ERK1/2 and JNK, and ERK1/2-mediated Egr1 expression. In SH-SY5Y cell monolayers and spheroids the exposure to NT-3 enhanced the apoptotic cascade triggered by VPA. Gene silencing of both TrkC-T1 and p75NTR prevented the NT-3 proapoptotic effect. Moreover, NT-3 enhanced p75NTR/TrkC-T1 co-immunoprecipitation. The results indicate that VPA upregulates TrkC by activating epigenetic mechanisms and signaling pathways, and sensitizes neuroblastoma cells to NT-3-induced apoptosis.

## 1. Introduction

The tropomyosin-related kinase receptor (Trk) C is a tyrosine kinase receptor that is preferentially activated by neurotrophin-3 (NT-3) [1]. Other structurally similar members of the neurotrophin receptor family are TrkA, which is the high affinity receptor for nerve growth factor (NGF), and TrkB, which binds brain derived neurotrophic factor (BDNF) and neurotrophin-4/5 [1]. Following the binding of NT-3, TrkC undergoes autophosphorylation and triggers the activation of distinct intracellular signaling pathways, including phosphatidylinositol 3-kinase (PI3K)/Akt, phospholipase C γ and MAP kinase pathways. These pathways are considered to play a key role in mediating the neurotrophin effects on neural cell proliferation, neuronal migration and differentiation, and synaptic organization and plasticity [1].

Although TrkC is encoded by a single *NTRK3* gene, it can be expressed in different isoforms produced through alternative splicing [2]. In addition to the tyrosine kinase containing full-length isoforms (TrkC-FL), TrkC displays a truncated isoform, termed TrkC-T1, which contains the extracellular and transmembrane portions of TrkC-FL, lacks the tyrosine kinase domain, and has a shorter intracellular domain including a unique 83 amino acid sequence [3]. As in the case of the truncated TrkB receptor, TrkC-T1 may negatively modulate the signaling of the TrkC-FL isoform either by sequestering the neurotrophin or forming inactive heterodimers [1]. Moreover, there is evidence that the binding of NT-3 to TrkC-T1 can signal independently of TrkC-FL to induce membrane ruffling and neuronal apoptosis [4,5,6,7]. NT-3 also binds with high affinity to the common neurotrophin p75NTR receptor [8], which belongs to the tumor necrosis factor receptor family [9]. p75NTR generally acts as a co-receptor for Trks, increasing their affinity and selectivity for the cognate neurotrophin [9]. However, when overexpressed or present in cells lacking Trks, p75NTR can promote apoptotic cell death [10,11]. Thus, NT-3 has the ability to induce different neurobiological outcomes depending on the predominant activation of a particular receptor molecule. 

Besides the crucial role in neurodevelopment, TrkC and the other Trks are known to be important players in affecting the growth and aggressiveness of pediatric tumors of neuronal origin [12]. In neuroblastoma (the most common solid extracranial tumor in children [13,14]) and medulloblastoma the expression of TrkC has been correlated with a good prognosis [15,16,17]. Similarly, enhanced TrkA expression was found to be associated with favorable neuroblastomas which may undergo spontaneous regression [18]. In neuroblastic primary tumors p75NTR transcript levels have been reported to be positively correlated with increased patient event-free and overall survival [19]. Moreover, p75NTR expression in neuroblastoma cells induces apoptosis [20]. Conversely, in primary neuroblastomas TrkB-FL expression has been found to be associated with *MYCN* gene amplification, an unfavorable prognostic marker [21], and its activation by BDNF has been shown to promote neuroblastoma cell survival, resistance to chemotherapy, anoikis and metastasis [22,23,24]. These observations suggest that the identification of pharmacological agents capable of changing the neurotrophin receptor profile of neuroblastoma cells into that of a less malignant phenotype may provide unique tools to counteract the growth of the tumor.

Preclinical studies have provided evidence that histone deacetylase (HDAC) inhibitors display anticancer activity against highly malignant tumors, including neuroblastoma [25,26,27]. In particular, valproic acid (VPA), an antiepileptic and mood stabilizer drug that preferentially inhibits HDAC of class I and IIa [27,28], has been shown to suppress the growth and viability of human neuroblastoma cells both in vitro and in vivo [29,30,31]. We have recently reported that VPA downregulates TrkB expression and signaling and impairs the prosurvival activity of BDNF in human neuroblastoma cells [32]. Moreover, VPA upregulates the expression of the p75NTR/sortilin receptor complex and promotes proNGF-induced neuroblastoma cell death [33]. We also observed that the exposure to VPA enhanced the expression of TrkC [32], but the mechanisms and the functional outcome of this change were not investigated.

In the present study, we examined the ability of VPA and other HDAC inhibitors to induce TrkC expression in different human neuroblastoma cell lines. Moreover, we investigated the intracellular pathways mediating the TrkC induction and the impact of this change on VPA inhibition of neuroblastoma cell viability.

## 2. Results

### 2.1. Induction of TrkC Expression by VPA in Human Neuroblastoma Cells

As indicated by Western blot analysis, prolonged exposure (24 h) to VPA (1 mM) increased the expression of the 140 kDa immunoreactive band of TrkC-FL and the 90 kDa band corresponding to TrKC-T1 in either *MYCN-*nonamplified SH-SY5Y cells or *MYCN-*amplified Kelly, BE(2)-C and IMR 32 cells (Figure 1A–D). Quantification of the VPA effect indicated that the increase was more marked in SH-SY5Y, Kelly and BE(2)-C (~20–30-fold increase) than IMR 32 cells (~2-fold increase), which displayed a higher basal receptor expression. In each neuroblastoma cell line, the VPA upregulation of the TrkC protein expression was accompanied by a comparable increase in the level of TrkC mRNA, as determined by qRT-PCR (Figure 1E–H).

### 2.2. VPA Enhances the Cell Surface Expression of TrkC and NT-3-Stimulated Intracellular Signaling

Analysis of cell surface protein expression in SH-SY5Y and BE(2)-C cells by using a cell-impermeable biotinylating agent indicated that in both neuroblastoma cell lines VPA (1 mM) increased the plasma membrane levels of TrkC to an extent similar to that observed in total cell lysates (Figure 2A,B). Immunofluorescence experiments performed by using nonpermeabilized SH-SY5Y cells and an antibody recognizing an extracellular domain of TrkC, to preferentially label the cell surface population of the receptor, showed that the percentage of cells positive for TrkC immunostaining was significantly enhanced in cells treated for 24 h with VPA (Figure 2C).

To investigate whether the VPA-induced increase in cell surface expression of TrkC was associated with an enhanced response to NT-3, we examined the ability of the neurotrophin to stimulate the phosphorylation of Akt and ERK1/2, two major mediators of the PI3K and MAP kinase signaling pathways, respectively, triggered by activated TrkC-FL [1]. As shown in Figure 2D,E, a brief exposure (5 min) to NT-3 (1 nM) failed to change the levels of either phospho-Akt or phospho-ERK1/2 in SH-SY5Y cells pretreated for 24 h with vehicle, consistent with the low expression of the receptor under basal conditions, but caused a significant increase in the phosphorylation state of both Akt and ERK1/2 in cells pretreated with VPA.

### 2.3. VPA-Induced TrkC Expression Is Mimicked by Other HDAC Inhibitors and Curtailed by Bromodomain Inhibition

We next investigated whether in human neuroblastoma cells the upregulation of TrkC could be induced by HDAC inhibitors other than VPA. As shown in Figure 3A–C, in SH-SY5Y, Kelly and BE(2)-C cells prolonged exposure (24 h) to entinostat (1 µM), a class I HDAC inhibitor, romidepsin (20 nM), a selective inhibitor of class I HDAC1 and 2, and the broad spectrum HDAC inhibitor vorinostat (200 nM) [27], elicited a significant increase in the levels of TrkC, although to a lesser extent than VPA. On the other hand, in IMR 32 cells only VPA induced a significant induction, whereas entinostat and vorinostat were without effect, and romidepsin caused a significant decrease in TrkC levels (Figure 3D). Exposure of SH-SY5Y cells to either the HDAC6 inhibitor tubacin (5 µM) or the HDAC8 inhibitor PCI-34051 (5 µM) failed to affect TrkC expression (Figure 3E). In agreement with previous findings [32,34,35], both tubacin and PCI-34051 had no effect on histone H3 acetylation (Figure 3E). On the other hand, under the same experimental conditions tubacin markedly increased the levels of acetylated α-tubulin, a major target of HDAC6 [34], demonstrating the efficacy of the treatment (Figure 3E). Analysis of concentration-response curves indicated that VPA increased TrkC expression and histone H3 acetylation with EC_50_ values of 0.54 ± 0.11 and 0.45 ± 0.13 mM, respectively (Figure 3F). The concentrations of VPA required to induce TrkC were within the range of the free drug concentrations reached with antitumor doses [36].

The bromo and extra terminal (BET) family of bromodomain (BRD) containing proteins regulate transcriptional changes at specific genes by recognizing acetylated lysine residues on histones H3 and H4 [37]. To explore the involvement of BET recruitment in VPA-induced TrkC upregulation, we employed the selective BET inhibitor (+)-JQ1 [37]. As shown in Figure 3G, pretreatment of SH-SY5Y cells with 250 and 500 nM (+)-JQ1 curtailed the VPA effect in a concentration-dependent manner.

### 2.4. The Upregulation of the Transcription Factor RUNX3 Contributes to VPA-Induced TrkC Expression

Overexpression of RUNX3, a member of the Runt family of transcription factors, has been shown to downregulate TrkB while promoting TrKC expression in developing dorsal root ganglia and in differentiated neuroblastoma cells [38,39]. We previously reported that in retinoic acid-differentiated SH-SY5Y cells VPA exposure upregulated RUNX3 to inhibit TrkB expression [32]. In the present study, we found that 24 h treatment of undifferentiated SH-SY5Y cells with VPA markedly increased the nuclear content of RUNX3 (Figure 4A). VPA treatment had no effect on the nuclear levels of two other transcription factors, AML1 and AP-2α, which have been shown to possess putative binding sites in the 5′ flanking region of the TrkC gene [40]. It has been reported that phosphorylation of AML1 at serine residues may enhance its transactivation ability [41]. Cell treatment with VPA failed to affect the nuclear levels of AML1 phosphorylated at Ser249 (Figure 4A).

To investigate whether the upregulation of RUNX3 mediated the VPA-induced increase of TrkC expression, SH-SY5Y cells were transfected with siRNA duplexes targeting RUNX3. This treatment suppressed the increase in RUNX3 protein levels almost completely and reduced the upregulation of TrkC elicited by VPA by 30–50%, as compared to control siRNA-treated cells (Figure 4B).

In human neuroblastoma cells RUNX3 gene transcriptional activity has been shown to be under negative control by EZH2, the histone methyltransferase catalytic subunit of the polycomb repressive complex 2 (PRC2) [42]. As shown in Figure 4C, treatment of SH-SY5Y cells with either deazaneplanocin A (5 µM), which causes cell depletion of EZH2 [43], or tazemetostat (1 µM), a direct inhibitor of EZH2 enzymatic activity [44], increased RUNX3 levels and enhanced TrkC expression.

### 2.5. Role of MAP Kinases in VPA-Induced TrkC Expression

VPA has been shown to regulate MAP kinases, such as ERK1/2 and JNK, in neuroblastoma and non-tumoral neuronal cells [45,46]. In agreement with these studies, time-course experiments showed that in SH-SY5Y cells the exposure to VPA (1 mM) stimulated the phosphorylation/activation of MEK1/2, the upstream activator of ERK1/2, ERK1/2, and the transcriptional activator Ets-like-1 (Elk-1), a major nuclear target of activated ERK1/2 [46] (Figure 5A–C). Elk-1 phosphorylation at serine/threonine sites, including Ser348, has been shown to be associated with an increase in the transcriptional activity [47]. The VPA stimulation of this signaling pathway became significant at 3 h and increased up to 24 h. VPA also induced a significant increase in the cellular levels of phospho-JNK, which started at 1 h, peaked at approximately 6 h and then slowly declined, remaining significant at 24 h (Figure 5D). The stimulation of JNK was associated with an early increase in phospho-c-Jun which was followed by enhancement of c-Jun levels, so that the phospho-c-Jun/c-Jun ratio rapidly returned to baseline levels (Figure 5E). Kinetic analysis of VPA induction of TrkC expression showed that this response occurred later than the activation of ERK1/2 and JNK, being evident only after 9 h of drug exposure (Figure 5F). The upregulation of TrkC persisted for at least 72 h of VPA treatment (Appendix A). As shown in Figure 5G,H, blockade of ERK1/2 signaling by the addition of the MEK1/2 inhibitor PD 98,059 (25 µM) reduced the VPA induction of TrkC by 65 ± 7%, whereas blockade of JNK with the selective inhibitor TCS JNK6o (10 µM) produced a minor, although significant, decrease of the VPA effect (42 ± 5% reduction). Conversely, cell pretreatment with the selective p38 MAPK inhibitor BIRB 0796 (1 µM), had no effect on the VPA response.

### 2.6. VPA Upregulates Egr1 to Induce TrkC Expression

The transcription factor early growth response 1 (Egr1) is under control of MAP kinases and is upregulated by activated Elk-1 [48]. Exposure of SH-SY5Y cells to VPA (1 mM) elevated Egr1 cellular levels at 6 h and produced a further increase at 24 h (Figure 6A). Both VPA and entinostat increased the nuclear content of Egr1 and decreased that of Sp1 (Figure 6B), a transcription factor which binds, like Egr1, to GC-rich consensus elements in the regulatory regions of a large number of genes [49]. Immunofluorescence analysis also showed that VPA treatment increased the percentage of cells expressing Egr1 and decreased that of cells positive for Sp1 (Figure 6C). The induction of Egr1 by either VPA or entinostat was almost completely suppressed by cell pretreatment with PD 98,059 (25 µM), implying the involvement of ERK1/2 signaling (Figure 6D,E). To investigate the role of Egr1 in the upregulation of TrkC, we first transfected SH-SY5Y cells with control and Egr1 siRNAs. As shown in Figure 6F, VPA greatly increased Egr1 levels in control siRNA-transfected cells, but had little effect in those transfected with Egr1 siRNA, indicating an efficient gene silencing. In cells transfected with Egr1 siRNA the induction of TrkC elicited by either VPA (1 mM) or entinostat (1 µM) was significantly smaller than that observed in control siRNA-treated cells (Figure 6G). We next examined the effects of cell transfection with Egr1 cDNA on TrkC induction by VPA. As shown in Figure 6H,I, this treatment enhanced basal Egr1 levels and significantly potentiated the VPA-induced increase in the expression of both Egr1 and TrkC.

### 2.7. NT-3 Potentiates Apoptosis in VPA-Treated Neuroblastoma Cells

Prolonged exposure of SH-SY5Y cells to VPA triggers apoptotic death [30,33]. It was therefore important to investigate whether NT-3 activation of the upregulated TrkC could affect the antineuroblastoma activity of VPA. To this aim, SH-SY5Y cells were preincubated for 24 h with either vehicle or VPA and then treated for an additional 24 h with either vehicle or NT-3. As shown in Figure 7A, immunofluorescence analysis of cleaved caspase 3 expression, a marker of apoptosis, showed that NT-3 (0.3 nM) had no effect in vehicle-pretreated cells, but significantly increased the percentage of apoptotic cells in VPA-pretreated samples. VPA-induced PARP cleavage, an event catalyzed by activated caspase 3, was also potentiated by the exposure to NT-3 (Figure 7B). In agreement with the potentiation of VPA-induced expression of apoptotic markers, cell treatment with the neurotrophin further decreased the viability of VPA-pretreated cells, while having no effect in vehicle-pretreated cells (Figure 7C). Under the same experimental conditions, NT-3 was found to enhance the VPA-induced phosphorylation of JNK and the increase in the cellular levels of the proapoptotic protein p53 (Figure 7D,E). Addition of TCS JNK6o (10 µM) before the exposure to NT-3 prevented the potentiation of p53 elevation and PARP cleavage elicited by NT-3 in VPA-pretreated cells (Figure 7F,G).

The ability of NT-3 to enhance VPA-induced apoptosis was also investigated in three-dimensional neuroblastoma cell cultures. We found that in spheroids generated from SH-SY5Y cells, prolonged incubation with VPA (1 mM) induced a significant upregulation of the expression of TrkC (Figure 7I). VPA or NT-3, added alone, had no significant effect on the growth of spheroids (Figure 7H). However, the exposure to NT-3 significantly inhibited the growth of spheroids pretreated with VPA. Western blot analysis of spheroid lysates indicated that the VPA treatment induced a significant increase in PARP cleavage which was potentiated by the addition of NT-3 (Figure 7J).

### 2.8. Role of TrkC-T1 and p75NTR in NT-3 Potentiation of VPA-Induced Neuroblastoma Cell Apoptosis

Both TrkC-T1 and p75NTR are known to promote cell death and are upregulated following exposure to VPA. Therefore, we investigated these two neurotrophin receptors as potential mediators of NT-3 proapoptotic activity. As shown in Figure 8A, cell transfection with a siRNA targeting TrkC-T1 markedly suppressed the VPA induction of TrkC-T1 while having little effect on the levels of TrkC-FL. The cell depletion of TrkC-T1 was associated with a loss of NT-3-induced potentiation of PARP cleavage in VPA-pretreated cells (Figure 8B). Cell transfection with p75NTR siRNA curtailed p75NTR expression and inhibited p75NTR upregulation by VPA (Figure 8C). This treatment prevented the enhancement of PARP cleavage by NT-3 (Figure 8D), indicating that both TrkC-T1 and p75NTR were required for the proapoptotic action of the neurotrophin. To investigate whether NT-3 promoted the interaction of TrkC-T1 with p75NTR, we performed immunoprecipitation experiments by using an antibody directed against TrkC-T1. As shown in Figure 8E, this antibody recognized TrkC-T1, but not TrkC-FL, and its immunoreactivity was almost completely abrogated by cell treatment with TrkC-T1 siRNA. Analysis of TrkC-T1 immunoprecipitates revealed the presence of p75NTR in cells pretreated with VPA but not vehicle (Figure 8F). In VPA-pretreated cells the exposure to NT-3 (1 nM) significantly enhanced the amount of p75NTR co-immunoprecipitated with TrkC-T1 (Figure 8F). No immunoreactive bands were detected when cell extracts were immunoprecipitated with preimmune IgG (results not shown).

## 3. Discussion

Previous studies have documented that VPA can adversely affect neuroblastoma cell growth and survival, but the molecular mechanisms contributing to these effects have not completely been defined. In the present study we show that in human neuroblastoma cells VPA upregulates the expression of the neurotrophin receptor TrkC and confers proapoptotic activity on the neurotrophin NT-3.

The upregulation of TrkC by VPA occurred at the protein and mRNA levels, was observed in cell lines with and without *MYCN* amplification and involved both the full-length and the truncated forms of the receptor. The extent of the upregulation appeared to vary among the cell lines investigated, being higher in those displaying low basal TrkC levels (SH-SY5Y, Kelly and BE(2)-C), than in those with higher basal receptor expression (IMR 32). This finding indicates that the effect of VPA is dependent on the cellular context.

The expression at the cell plasma membrane is a critical condition for the Trk receptor to be responsive to the cognate neurotrophin [1]. Previously, we have reported that in retinoic acid-differentiated SH-SY5Y cells the downregulation of TrkB induced by VPA translated into a reduced BDNF intracellular signaling [32]. In the present study, we show that in SH-SY5Y and BE(2)-C cells VPA enhanced the cell surface expression of both TrkC-FL and TrkC-T1 isoforms. This effect was associated with enhanced Akt and Erk1/2 activation by NT-3, indicating the occurrence of a functional coupling of upregulated TrkC-FL to downstream signaling pathways.

Like VPA, entinostat, romidepsin and vorinostat induced TrkC expression in SH-SY5Y, Kelly and BE(2)-C cells, showing that this response could be triggered by HDAC inhibitors with different chemical structure and pharmacological properties. However, their efficacy appeared consistently lower than that of VPA, and in IMR 32 cells only VPA was able to upregulate TrkC, whereas romidepsin decreased the receptor expression. Considering that these agents have overlapping actions on HDACs, the precise reasons for their different behavior are not clear. In IMR 32 cells, entinostat, romidepsin and vorinostat were found to be effective inducers of p75NTR as well as VPA (Appendix A), indicating that the cells were able to respond to these HDAC inhibitors as observed in other neuroblastoma cell lines [33]. Thus, it is possible that some unique properties of VPA not shared by the other HDAC inhibitors may underlie the superior efficacy of the drug in upregulating TrkC.

The finding that the HDAC6 inhibitor tubacin and the HDAC8 inhibitor PCI-34051 did not affect TrkC expression and histone acetylation suggests that the TrkC upregulation involves the inhibition of specific HDAC classes capable of chromatin remodeling. The observation that VPA induced TrkC expression and histone H3 acetylation with similar potencies, and the finding that the BET inhibitor (+)-JQ1 curtailed the TrkC upregulation, are also in line with the idea that epigenetic mechanisms involving HDAC inhibition and histone acetylation mediated VPA induction of TrkC.

One epigenetic mechanism found to mediate TrkC upregulation by VPA is the induction of RUNX3. In SH-SY5Y cells, gene silencing of RUNX3 by siRNA treatment effectively counteracted the enhancement of RUNX3 and attenuated the induction of TrkC by VPA. Previously, we observed that the upregulation of RUNX3 by VPA was associated with decreased expression of EZH2 [32], a negative regulator of the RUNX3 gene [42]. Moreover, cell treatment with 3-deazaneplanocin A and tazemetostat, which deplete and inhibit EZH2, respectively, derepressed RUNX3 expression and increased TrkC levels. These results support the participation of the EZH2-RUNX3 epigenetic pathway in TrkC induction by VPA. However, it is noteworthy that RUNX3 gene silencing reduced the VPA upregulation of TrkC only partially and that the increase of TrkC expression elicited by either deazaneplanocin A or tazemetostat was modest as compared to their stimulatory effects on RUNX3 levels. These discrepancies suggest that the derepression of RUNX3 might not be the sole mechanism responsible for the upregulation of TrkC.

Activation of MAP kinases appeared to provide a relevant contribution to the induction of TrkC. In SH-SY5Y cells, the activation of ERK1/2 and JNK by VPA preceded the increase in TrkC and pharmacological blockade of either ERK1/2 or JNK inhibited VPA-induced TrkC upregulation, with a greater reduction obtained with suppression of ERK1/2. Like ERK1/2, JNK has been demonstrated to phosphorylate and activate Elk-1 [50]. Therefore, one possible mechanism by which JNK contributes to TrkC induction is the activation of the same transcriptional events regulated by ERK1/2. On the other hand, inhibition of p38 MAPK had no effect, consistent with the reported lack of activation of this kinase following 24 h exposure of SH-SY5Y cells to VPA [45].

Downstream of ERK1/2, we identified Egr1 as a putative transcription factor mediating TrkC upregulation. Treatment of SH-SY5Y cells with either VPA or entinostat increased the total cell and nuclear content of Egr1 in a manner that was dependent on ERK1/2 signaling. Cell depletion of Egr1 curtailed TrkC upregulation by either VPA or entinostat, whereas Egr1 overexpression potentiated the stimulatory effect of VPA. The induction of Egr1 observed following treatment with either VPA or entinostat was accompanied by a reduced nuclear content of Sp1. While the mechanisms involved in Sp1 downregulation by the HDAC inhibitors remain to be determined, it is noteworthy that Egr1 and Sp1 have been found to compete for the binding to GC-rich consensus elements in the regulatory regions of different genes [51,52]. As the 5′ flanking region of the *NTRK3* gene contains several GC-rich sequences [40], one may speculate that the opposing effects on Egr1 and Sp1 expression induced by HDAC inhibitors may favor the transactivating activity of Egr1 at this gene. Interestingly, in human neuroblastoma cells Sp1 has been reported to form a repression complex with MYCN and MIZ1, and that Sp1 cell depletion has been found to induce the expression of p75NTR and TrkA [53]. Additional studies are required to investigate whether Sp1 also acts as a negative regulator of the *NTRK3* gene.

An important issue which remains to be addressed is how VPA activates ERK1/2 and JNK in human neuroblastoma cells. Previous studies in other cell types have found no correlation between VPA-induced ERK1/2 phosphorylation and histone acetylation, indicating that MAP kinase activation occurs independently of HDAC inhibition [54,55]. However, whether VPA can directly control the activity of MAP kinases by affecting their acetylation state has not been investigated.

NT-3 is a crucial factor controlling neural development, survival and differentiation [56]. While in medulloblastoma NT-3 has been found to induce apoptotic cell death [57], in human neuroblastoma cells the production of this neurotrophin has been shown to promote survival by blocking the apoptosis triggered by proteolytic processing of TrkC intracellular domain [58]. Having observed that VPA treatment promoted NT-3-induced Akt and ERK1/2 activation, which are known to mediate anti-apoptotic signals [1], we expected that the upregulation of TrkC would be accompanied by a prosurvival outcome following NT-3 treatment. In contrast, we found that NT-3 potentiated the proapoptotic cascade stimulated by VPA, as indicated by enhanced active caspase 3 formation, PARP cleavage, and cell death in response to VPA. These changes were associated with a greater increase of JNK activation and p53 accumulation when NT-3 was added to VPA-pretreated cells. Collectively, these results suggest that VPA treatment leads to a condition in which NT-3-induced death signaling overcomes the stimulation of prosurvival pathways. In a variety of cancer cells HDAC inhibitors have been shown to increase p53 levels by promoting its acetylation, which protects the protein from ubiquitination [27,59]. In addition, activation of JNK signaling has been shown to stabilize and activate p53 and to enhance p53-dependent apoptosis [60]. In agreement with this finding, pharmacological blockade of JNK in cells pretreated with VPA prevented NT-3-induced potentiation of p53 accumulation and PARP cleavage.

Three-dimensional cell cultures of cancer cells as spheroids are considered to constitute a more representative model of solid tumors than two-dimensional cultures as they allow intercellular interactions and the creation of a tumor microenvironment [61]. Moreover, it has been reported that in neuroblastoma cells the expression of tumor proteins and the sensitivity to anticancer drugs may be different in three- vs. two-dimensional systems [62]. It was therefore important to investigate whether VPA affected TrkC expression and promoted NT-3-dependent cytotoxicity in a three-dimensional cell culture. We found that the ability of VPA to upregulate TrkC and to sensitize the cells to the proapoptotic actions of NT-3 was retained in spheroids generated from SH-SY5Y cells, raising the possibility that these effects also occur under in vivo conditions.

Gene silencing experiments indicated that in VPA-treated SH-SY5Y cells the upregulation of both TrkC-T1 and p75NTR created a crucial condition allowing NT-3 proapoptotic effects. TrkC-T1 induction may favor the receptor association with the scaffold protein tamalin [5,63], which can stimulate the Rac1 GTPase, an upstream activator of JNK [64]. Activation of JNK and p53 accumulation may also be triggered by NT-3 binding to the upregulated p75NTR, as it has been observed in neurons undergoing neurotrophin-induced apoptotic death [10,65]. Thus, the combined action of NT-3 at TrkC-T1 and p75NTR may underlie the potentiation of JNK activation and apoptosis in VPA-treated SH-SY5Y cells. Accordingly, we found that in these cells NT-3 enhanced the amount of p75NTR co-immunoprecipitated with TrkC-T1, indicating that the neurotrophin promoted the physical interaction of the two receptors. It is possible that the formation of a receptor complex and the cooperation of TrkC-T1 and p75NTR are necessary for NT-3 generation of proapoptotic signals. A functional interaction between TrkC-T1 and p75NTR has previously been observed in neural crest cells, where both receptors were required to promote NT-3 stimulation of cell differentiation [66].

In conclusion, the present study demonstrates that, by activating epigenetic mechanisms involving derepression of RUNX3 expression and stimulating signaling pathways mediated by ERK1/2 and JNK, VPA upregulates TrkC in human neuroblastoma cells (Figure 9). Moreover, the study shows that the enhanced expression of TrkC-T1 and p75NTR promoted by VPA sensitizes neuroblastoma cells to NT-3-induced apoptosis. This novel finding, together with the previous observations of downregulation of TrkB and induction of p75NTR/sortilin receptor complex, further supports the idea that changes in the neurotrophin receptor expression and functional responses contribute to the antineuroblastoma activity of VPA.

## 4. Materials and Methods

### 4.1. Materials

Recombinant human NT-3 was obtained from ImmunoTools (Friesoythe, Germany). Entinostat and PD 98,059 were obtained from Santa Cruz Biotechnology (Dallas, TX, USA). Vorinostat, romidepsin, PCI-34051, tubacin, 3-deazaneplanocin A hydrochloride, (+)-JQ-1 and tazemetostat were obtained from MedChem Express Europe (Sollentuna, Sweden). VPA and 4′,6-diamidino-2phenylindole dihydrochloride (DAPI) were obtained from Sigma-Aldrich (St. Louis, MO, USA). BIRB 0796 was obtained from Axon Medchem BV (Groningen, The Netherlands), whereas TCS JNK6o was from Tocris Bioscience (Bristol, UK).

### 4.2. Cell Culture

Human neuroblastoma cell lines SH-SY5Y, BE(2)-C and IMR 32 were obtained from the European Collection of Authenticated Cell Cultures (Salisbury, UK), whereas the human neuroblastoma cell line Kelly was from CLS Cell Lines Service GmbH (Eppelheim, Germany). The cell lines were authenticated by the vendors. SH-SY5Y and BE(2)-C cells were grown in Ham’s F12/MEM medium (1:1) (Sigma-Aldrich) containing 2 mM L-glutamine (Sigma-Aldrich) and 1% non-essential amino acids (NEAA) (Sigma-Aldrich). IMR 32 cells were grown in MEM medium (Sigma-Aldrich) containing 2 mM L-glutamine and 1% NEAA. Kelly cells were cultured in RPMI 1640 containing 2 mM L-glutamine (Sigma-Aldrich). Culture media were supplemented with 10% fetal calf serum (FCS) and 100 U/mL penicillin-100 µg/mL streptomycin (Sigma-Aldrich). Cells were maintained at 37 °C in a humidified atmosphere of 5% CO_2_ in air. Sub-confluent cultures were split every 72 h and seeded at the density of 1–3 × 10^4^/cm^2^ using 0.25% trypsin/EDTA (Sigma-Aldrich). After resuscitation, cells were used for no more than 10–15 passages. Cells were periodically checked for mycoplasma contamination by using the MycoFluor Mycoplasma Detection kit (Invitrogen-Life Technologies).

### 4.3. Cell Treatment

Unless otherwise specified, neuroblastoma cells were washed with phosphate buffered saline (PBS) and incubated in medium containing 1% FCS. Cells were treated with the test agents as indicated in the text, and maintained at 37 °C in a humidified atmosphere of 5% CO_2_ in air. Control samples received an equal amount of vehicle. Cell lysates were prepared by washing with PBS and scraping the cells into an ice-cold lysis buffer containing PBS, 0.1% sodium dodecyl sulphate (SDS), 1% Nonidet P-40, 0.5% sodium deoxycholate, 2 mM EDTA, 2 mM EGTA, 4 mM sodium pyrophosphate, 2 mM sodium orthovanadate, 10 mM sodium fluoride, 20 nM okadaic acid, 1 mM phenylmethylsulphonyl fluoride (PMSF), 0.5% phosphatase inhibitor cocktail 3 and 1% protease inhibitor cocktail (Sigma-Aldrich) (RIPA buffer). The samples were sonicated for 5 s in an ice bath and aliquots of cell extracts were taken for protein determination by the Bio-Rad protein assay (Bio-Rad Lab, Hercules, CA, USA).

### 4.4. Cell Transfections

To knock down gene expression, SH-SY5Y cells were transfected with 50 pmol/mL of one of the following small interfering RNA (siRNA) duplexes: control (non-targeting) siRNA-A (sc-37007), RUNX3 siRNA (sc-37679), Egr1 siRNA (sc-29303), p75NTR siRNA (sc-36051) (Santa Cruz Biotechnology) or TrkC-T1 siRNA (custom synthetized by Ambion-Life Technologies according to published sequence [67]). Lipofectamine RNAiMAX (Invitrogen) was used as the transfection reagent. Cells grown in 6-well plates at ~70% confluency were incubated in antibiotic-free medium for 24 h. The medium was renewed and the cells were incubated with siRNA duplexes for 5–6 h at 37 °C. Thereafter, the medium was replaced by the growth medium and the cells were analyzed 48 h post-transfection. The transfection efficiency (50–65%) was determined in parallel samples incubated with 50 pmol/mL of fluorescein-conjugated control siRNA-A (sc-36869, Santa Cruz Biotechnology).

To induce Egr1 overexpression, subconfluent SH-SY5Y cell cultures maintained in antibiotic-free medium for 24 h were incubated in Opti-MEM 1 reduced serum medium (Invitrogen) and transfected with 1 µg of either human Egr1 cDNA (GenScript, Piscataway, NJ, USA) or pcDNA3.1+ empty vector by using Lipofectamine 2000 (Invitrogen) as the transfection agent. After 12 h, the medium was removed and replaced by fresh growth medium. Cells were used 48 h post-transfection.

### 4.5. Quantitative Reverse Transcription Polymerase Chain Reaction (qRT-PCR)

Cells were incubated in a medium containing 1% FCS and treated for 24 h with either vehicle or VPA (1 mM). Thereafter, cells were washed and total RNA was isolated by using TRIzol reagent (Invitrogen) and the PureLink RNA mini kit (Ambion- Life Technologies). Thereafter, samples were subjected to a Turbo DNase (Ambion-Life Technologies) digestion. The purity and quantity of the isolated RNA were determined by UV absorbance at 260 and 280 nm. First-strand cDNA synthesis was performed using 2 µg of total RNA using the SuperScript VILO cDNA synthesis kit (Invitrogen). Two-hundred ng of cDNA for reaction was used for quantitative real time PCR amplification with SYBR Green PCR Master Mix (Applied Biosystems). The PCRs were carried out on a Real-Time PCR System (StepOne, Applied Biosystems) under the following conditions: an initial holding stage at 95 °C for 10 min was followed by 45 cycles, denaturation at 95 °C for 15 s, primer annealing and extension at 60 °C for 1 min, and a dissociation curve to the end of a real time run (melt curve 95 °C for 15 s, 60 °C for 1 min and 95 °C for 15 s). PCR primers used were: human TrkC (NCBI accession no. NM_002530) forward CCGACACTGTGGTCATTGGCAT, reverse CAGTTCTCGCTTCAGCACGATG; human β-actin (NCBI accession no. NM_0001101.3) forward AGCCTCGCCTTTGCCGATCCG, reverse CATGCCGGAGCCGTTGTCGAC. The comparative Ct values method was used to calculate the relative quantity TrkC expression.

### 4.6. Biotinylation of Cell Surface Proteins

Surface biotinylation of cell proteins was performed as previously described [68]. Briefly, SH-SY5Y and BE(2)-C cells treated with either vehicle or VPA for 24 h were incubated for 45 min at 4 °C with the cell impermeable biotinylating agent sulfosuccinimidyl-6-(biotin-amido)hexanoate (sulpho-NHS-LC-biotin) (0.50 mg/mL) (Pierce, Rockford, IL, USA). Thereafter, the cells were washed with PBS containing 20 mM glycine and solubilized by incubation in RIPA buffer supplemented with 1% Triton X-100. Cell extracts were centrifuged at 14,000× *g* for 5 min at 4 °C and the supernatants incubated overnight at 4 °C with streptavidin-conjugated agarose beads. Following washing, the beads were mixed with sample buffer and incubated for 2 min at 100 °C. The proteins were then analyzed by Western blot.

### 4.7. Isolation of Cell Nuclei

Cells grown to confluency in 100 mm dishes were incubated with the test agents as indicated, washed with ice-cold PBS (pH 7.4) and scraped in ice-cold lysis buffer containing PBS, 10 mM Tris-HCl, 2 mM MgCl_2_, 10 mM NaCl, 0.5 mM EGTA, 2 mM sodium orthovanadate, 10 mM sodium fluoride, 1 mM PMSF, 0.05% Nonidet, 0.5% phosphatase inhibitor cocktail 3 and 1% protease inhibitor cocktail (pH 7.4). Following centrifugation at 3000× *g* for 10 min at 4 °C, the supernatant was collected and centrifuged at 24,000× *g* for 20 min. The supernatant was used as the cytosolic fraction. The pellet of the first centrifugation was washed three times with an ice-cold buffer containing 10 mM PIPES-KOH, 300 mM sucrose, 2 mM MgCl_2_, 10 mM NaCl, 0.5 mM EGTA, 2 mM sodium orthovanadate, 10 mM sodium fluoride, 1 mM PMSF, 0.5% phosphatase inhibitor cocktail 3 and 1% protease inhibitor cocktail (pH 7.4), and layered over a 1 mL cushion of 1 M sucrose, 2 mM sodium orthovanadate, 10 mM sodium fluoride, 1 mM PMSF, 0.5% phosphatase inhibitor cocktail 3 and 1% protease inhibitor cocktail. Following centrifugation at 3000× *g* for 10 min at 4 °C, the nuclei present in the pellet were washed and the proteins were extracted by incubation on ice for 30 min in a buffer containing 20 mM HEPES-NaOH, 300 mM NaCl, 2 mM MgCl_2_, 0.2 mM EDTA, 2 mM sodium orthovanadate, 10 mM sodium fluoride, 1 mM PMSF, 0.5% phosphatase inhibitor cocktail 3 and 1% protease inhibitor cocktail (pH 7.9). The samples were centrifuged at 24,000× g for 10 min at 4°C and the nuclear extracts were mixed with sample buffer and analyzed by Western blot.

### 4.8. Immunoprecipitation

SH-SY5Y cells treated for 24 h with either vehicle or 1 mM VPA were incubated with either vehicle or 1 nM NT-3 for 1 h. The cells were lysed with ice-cold RIPA buffer supplemented with 1% Triton × 100 and incubated for 30 min at ice-bath temperature. Following centrifugation at 10,000× *g* for 10 min at 4 °C, the supernatant (~500 µg of protein) was incubated overnight at 4 °C with either anti-TrkC-T1 antibody (1:100) (cat no. 600-401-993, Rockland, Limerick, PA, USA) or preimmune rabbit IgG (1:100) (Santa Cruz Biotechnology). Thereafter, 50 µL of Pure Proteome Protein G magnetic beads (Millipore, Burlington, MA, USA) were added and samples incubated at 4 °C for 3 h with continuous rotation at 4 °C. The beads were washed 5 times with ice-cold PBS/0.1% Tween 20 buffer. After the last wash, the beads were resuspended in 2× sample buffer and boiled for 5 min.

### 4.9. Western Blot Analysis

Cell proteins were separated by SDS-polyacrylamide gel electrophoresis and electrophoretically transferred to polyvinylidene difluoride membranes (Amersham Biosciences, Piscataway, NJ, USA) by using a semi-dry apparatus. Membranes were blocked with 5% low-fat dry milk, washed and incubated overnight at 4 °C with one of the following primary antibodies: TrkC (cat. no. 3376, Cell Signaling Technology) (1:1000), TrkC-T1 (cat no. 600-401-993, Rockland, Limerick, PA, USA) (1:1000), p75NTR (cat no. 8238, Cell Signaling Technology) (1:1000), Jun N-terminal kinase (JNK) (sc-571, Santa Cruz Biotechnology) (1:2000), phospho-JNK (Thr183/Tyr185) (cat. no. 9912, Cell Signaling Technology) (1:1000), phospho-c-Jun (Ser73) (cat. no. 3270, Cell Signaling Technology) (1:1000), c-Jun (cat. no. 9165, Cell Signaling Technology) (1:1000), phospho-Akt (Thr308) (cat. no. 2965, Cell Signaling Technology) (1:5000), Akt1/2/3 (sc-8312, Santa Cruz Biotechnology) (1:1000), extracellular signal-regulated kinases 1 and 2 (ERK1/2) (cat no. 9102, Cell Signaling Technology), phospho-ERK1 (Thr202/Tyr204)/ERK2 (Thr185/Tyr187) (cat no. RA15002, Neuromics, Nothfield, MN, USA) (1:5000), phospho-MEK1/2 (Ser217/221) (cat no. 9154, Cell Signaling Technology) (1:1000), MEK1/2 (sc-81504, Santa Cruz Biotechnology) (1:1000), phospho-Elk-1 (Ser383) (cat. no. 9181, Cell Signaling Technology) (1:1000), Elk-1 (sc-365876, Santa Cruz Biotechnology) (1:1000), RUNX3/AML2 (RUNX3) (cat no. 9647, Cell Signaling Technology) (1:1000), RUNX1/AML1 (AML1) (cat. no. 4336, Cell Signaling Technology) (1:1000), phospho-AML1 (Ser249) (cat. no. 4327, Cell Signaling Technology) (1:1000), AP-2α (cat. 3215, Cell Signaling Technology) (1:1000), Egr1 (cat. no. 4154, Cell Signaling Technology) (1:1000), Sp1 (cat. no. 9389, Cell Signaling Technology) (1:1000), p53 (sc-6243, Santa Cruz Biotechnology) (1:500), HDAC1 (sc-81598, Santa Cruz Biotechnology) (1:2000), poly(ADP-ribose)polymerase (PARP) (cat. no. 9542, Cell Signaling Technology) (1:1000), pan-cadherin (cat no. 4073, Cell Signaling Technology) (1:2000); acetylated tubulin (cat. no. T7451, Sigma-Aldrich) (1:1000), α-tubulin (sc-5286, Santa Cruz Biotechnology) (1:1000), histone H3 (acetyl Lys9, Lys14) (cat no. GTX122648, GeneTex Inc., Irvine, CA, USA) (1:1000), histone H3 (cat. no. GTX 122148, GeneTex Inc.) (1:2000), actin (cat. no. A2066, Sigma-Aldrich) (1:2000), glyceradheyde-3-phosphate-dehydrogenase (GAPDH) (cat no. 247002, Synaptic Systems GmbH, Gottingen, Germany) (1:10,000). Thereafter, the membranes were washed and incubated with an appropriate horseradish peroxidase-conjugated secondary antibody (Santa Cruz Biotechnology). Immunoreactive bands were detected by using Clarity Western ECL substrate (Bio-Rad Lab.) and digital images were obtained by using either ECL Hyperfilm (Amersham) with Image Scanner III (GE Healthcare, Milan, Italy) or Luminescence Image analyzer LAS 4000 (FujiFilm, Tokyo, Japan). Band densities were determined using the NIH ImageJ software (US National Institutes of Health, Bethesda, MA, USA). The optical density of the phosphorylated protein bands was normalized to the density of the corresponding total protein in the same sample. For analysis of PARP, the formation of the cleaved protein was normalized to the level of the corresponding uncleaved PARP measured in the same sample. For the remaining proteins, the densitometric values were normalized to the levels of either actin, GAPDH, or subcellular fraction marker, as indicated. The size of immunoreactive bands was determined by using molecular weight standards detected with an ECL suitable antibody (sc-2035, Santa Cruz Biotechnology) (1:1000).

### 4.10. Immunofluorescence Analysis

Cells grown onto coverslips coated with poly-L-lysine (Sigma-Aldrich) were exposed to the test agents as indicated in the text, washed, and fixed in 4% paraformaldehyde. For analysis of TrkC expression, nonpermeabilized cells were blocked with 3% BSA plus 1% normal goat serum and incubated overnight with an antibody recognizing the extracellular domain of TrkC (cat. no. 3376, Cell Signaling Technology) (1:50). For the detection of cleaved caspase 3, Egr1 and Sp1 immunoreactivities, cells were permeabilized with 0.2% Triton X-100, blocked and incubated overnight with either anti-cleaved caspase 3 (cat no. 9661, Cell Signaling Technology) (1:200), anti-Egr1 (cat. no. 4154, Cell Signaling Technology) (1:200), or anti-Sp1 (cat. no. 9389, Cell Signaling Technology) (1:200). Control samples were incubated in the presence of rabbit pre-immune IgG. Thereafter, cells were incubated with Alexa-Fluor488-conjugated secondary antibody (1:1500) (Invitrogen-Molecular Probes) and cell nuclei were stained with 0.1 µg/mL DAPI. Cells were analyzed with an Olympus BX61 microscope equipped with a F-View II CCD-camera by using either 40× or 60× objective lens. Digital images were acquired using constant camera settings within each experiment and were analyzed using the program Cell P (Olympus Soft Imaging Solutions, Homburg, Germany). At least 10 fields were randomly selected for each sample and only cells showing an unobstructed nucleus or soma were considered.

For quantification of TrkC, cleaved caspase 3, Egr1, and Sp1 expression, the average pixel intensity was measured within the region of the cell soma or the nucleus, as appropriate, and in an adjacent area, which was used as background value. Cells were deemed to be positive if the average pixel intensity was equal or above a threshold value corresponding to one standard deviation above the average pixel intensity of the respective control samples. No labeling was detected in samples treated with preimmune IgG. Images were analyzed by an investigator unaware of the treatment.

### 4.11. Assay of Cell Viability

A luminescence analysis using the Real Time-GLO MT assay kit (Promega, Madison, WI, USA) was used to determine cell viability. Cells grown in 96-well plates (ViewPlate, PerkinElmer) were exposed to the test agents as indicated in the text and then incubated with the reagents provided by the kit following the manufacturer’s instructions. Luminescence intensity was measured by using a Wallac Victor III microplate reader (PerkinElmer). Assays were performed in triplicate.

### 4.12. Tumor Spheroid Generation and Analysis

To generate spheroids, SH-SY5Y cells were seeded in a 96-well U-bottom ultra-low attachment plate (Nunclon Sphera, Thermo Fisher Scientific) at the density of 3 × 10^3^ cells/well in 200 µL of complete growth medium. The plates were centrifuged at 200× *g* for 5 min at room temperature to generate one spheroid in each individual well and placed in an incubator at 37 °C and 5% CO_2_. Spheroids were incubated with the test agents as specified in the text 24 h after cell seeding. Spheroids were examined by light microscopy using an Olympus IX 51 inverted microscope equipped with a 10× Plan achromatic objective and an Olympus digital camera. At the end of the treatment, images were acquired and the area of each spheroid was measured by using the ImageJ software by an investigator unaware of the experimental design. Six individual spheroids were examined for each experimental group. For Western blot analysis, four spheroids from the same experimental group were pooled, centrifuged, resuspended in ice-cold RIPA buffer and lysed by sonication. Aliquots of the lysates containing an equal amount of protein were mixed with sample buffer, heated at 100 °C and subjected to SDS-PAGE.

### 4.13. Statistical Analysis

Results are reported as the mean ± SD. Statistical analysis was performed by using the program GraphPad Prism 9 (San Diego, CA, USA). Unless otherwise indicated, data are expressed as percentage or fold stimulation of control, which was included in each independent experiment. The control group was set as 100 or 1 with a variance obtained by expressing each control value as a percentage of the mean of the raw values of the control group. In the experiments where control values were equal to zero, values of experimental groups were expressed as a percentage of the maximal effect set as 100. The variance of this value was determined in the same manner as for the control group. The normal distribution of the data was assessed by Shapiro-Wilk test. Statistical analysis was performed by using either the unpaired Student’s *t* test for comparing two groups, or one-way analysis of variance (ANOVA) followed by Tukey’s test when comparing multiple groups. A value of *p* < 0.05 was considered to be statistically significant.

## Figures and Tables

**Figure 1 ijms-22-07790-f001:**
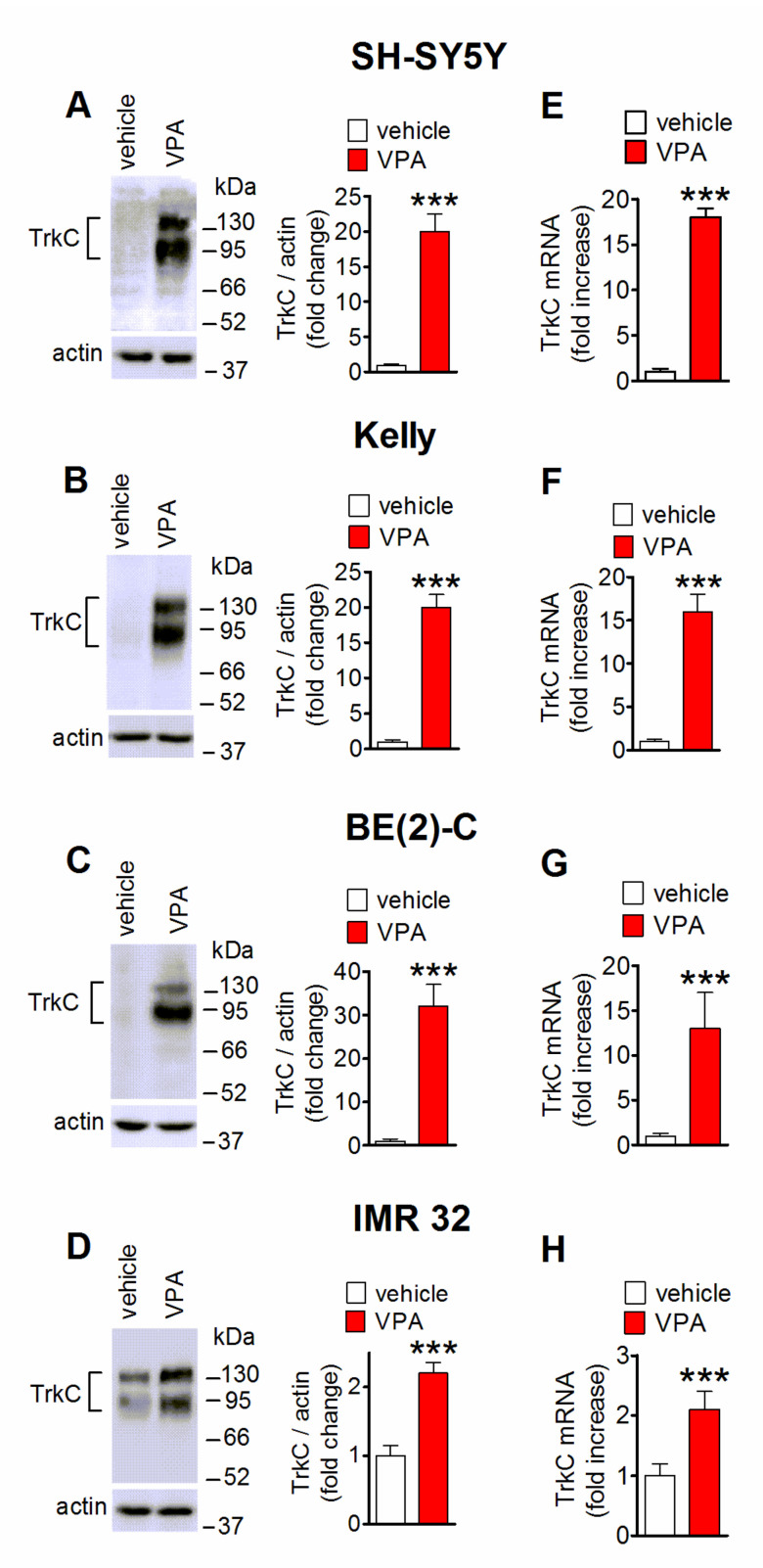
VPA upregulates TrkC expression in human neuroblastoma cell lines. (**A**–**D**) Cells were treated with either vehicle or 1 mM VPA for 24 h and then analyzed for TrkC protein levels by Western blot. Values are the mean ± SD of five (**A**) and four (**B**–**D**) independent experiments. The position of molecular mass standards is indicated on the right side of each blot. (**E**–**H**) Quantitative real-time reverse transcription polymerase chain reaction (qRT-PCR) analysis of TrkC mRNA in cells treated for 24 h with either vehicle or 1 mM VPA. Values are the mean ± SD of four independent experiments. *** *p* < 0.001 vs. control (vehicle-treated cells) by Student’s *t* test.

**Figure 2 ijms-22-07790-f002:**
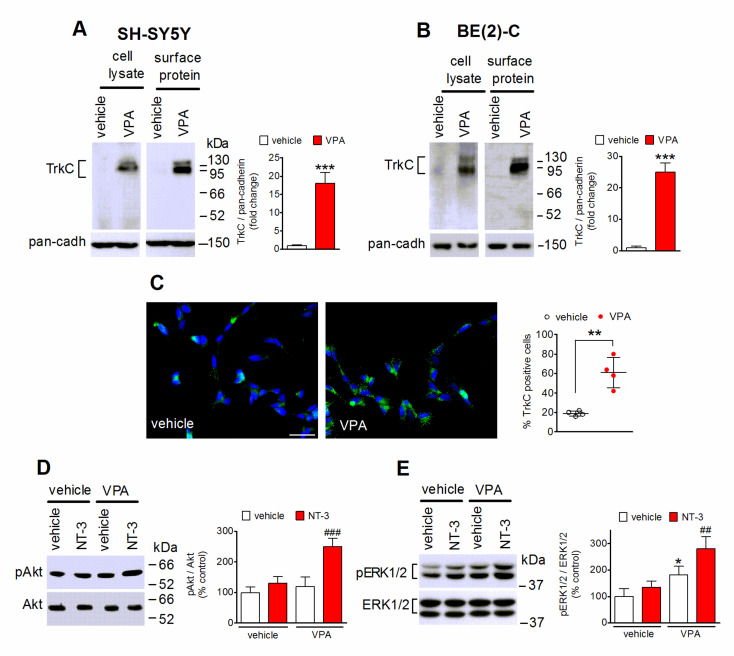
VPA increases cell surface expression of TrkC. (**A**,**B**) SH-SY5Y and BE(2)-C cells were incubated for 24 h with either vehicle or 1 mM VPA. Cells were then treated with the cell impermeant biotinylating agent sulfosuccinimidyl-6-(biotin-amido) hexanoate and solubilized proteins were isolated by precipitation with streptavidin-conjugated agarose beads. The total extract (cell lysate) and precipitated proteins (surface protein) were analyzed for TrkC by Western blot. The bar graphs show the changes in cell surface TrkC levels normalized to pan-cadherin (pan-cadh) levels, used as plasma membrane marker. Values are the mean ± SD of four independent experiments. (**C**) SH-SY5Y cells grown onto glass coverslips and treated for 24 h with either vehicle or 1 mM VPA were analyzed for TrkC expression by immunofluorescence without permeabilization and with an antibody recognizing an extracellular domain of the receptor (green color). Nuclei were stained in blue with 4′,6-diamidino-2phenylindole dihydrochloride (DAPI). Scale bar, 25 µm. Values reported in the scatterplot are the mean ± SD of four independent experiments. ** *p* < 0.01, *** *p* < 0.001 vs. control (vehicle) by Student’s *t* test. (**D**,**E**) SH-SY5Y cells were incubated in serum-free medium with either vehicle or 1 mM VPA for 24 h and then exposed for 5 min to either vehicle or 1 nM NT-3. Cell lysates were analyzed for the expression of phospho-Thr308-Akt (pAkt) and total Akt (**D**) and phospho-ERK1/2 (pERK1/2) and total ERK1/2 (**E**). Values are the mean ± SD of four independent experiments. * *p* < 0.05 vs. control (vehicle + vehicle); ^##^
*p* < 0.01, ^###^
*p* < 0.001 vs. VPA + vehicle by one-way analysis of variance (ANOVA) followed by Tukey’s test.

**Figure 3 ijms-22-07790-f003:**
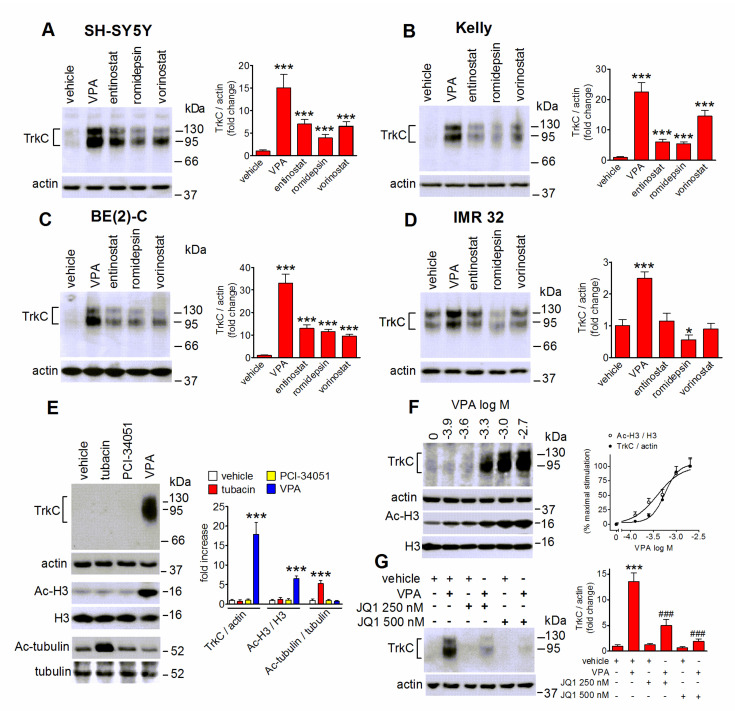
Effects of HDAC and BET inhibitors on TrkC expression in human neuroblastoma cells. (**A**–**D**) Cells were incubated for 24 h with either vehicle, 1 mM VPA, 1 µM entinostat, 20 nM romidepsin or 200 nM vorinostat. Cell lysates were analyzed for TrkC expression by Western blot. Values are the mean ± SD of four independent experiments. * *p* < 0.05 *** *p* < 0.001 vs control (vehicle). (**E**) SH-SY5Y cells were treated for 24 h with either vehicle, 5 µM tubacin, 5 µM PCI-34051 or 1 mM VPA. Cell lysates were analyzed for TrkC, actin, acetyl-Lys9/Lys14 histone H3 (Ac-H3), H3, acetyl-tubulin (Ac-tubulin) and tubulin. Densitometric ratios are reported as fold increase with respect to control (vehicle) and are the mean ± SD of four independent experiments. *** *p* < 0.001 vs the corresponding control (vehicle). (**F**) SH-SY5Y cells were incubated for 24 h with the indicated concentrations of VPA. Cell lysates were analyzed for TrkC expression and histone H3 acetylation. Values are the mean ± SD of four independent experiments. (**G**) SH-SY5Y cells were preincubated for 1 h with either vehicle, 250 nM or 500 nM (+)-JQ1 (JQ1) and then treated with either vehicle or 1 mM VPA for 24 h. Cell lysates were analyzed for TrkC expression. Values are the mean ± SD of four independent experiments. *** *p* < 0.001 vs. control (vehicle + vehicle); ^###^
*p* < 0.001 vs. vehicle + VPA by ANOVA followed by Tukey’s test.

**Figure 4 ijms-22-07790-f004:**
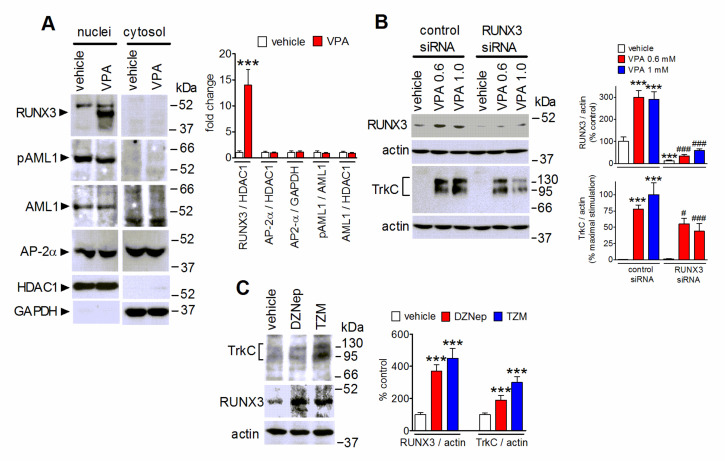
Involvement of RUNX3 in VPA-induced upregulation of TrkC. (**A**) SH-SY5Y cells were treated for 24 h with either vehicle or 1 mM VPA. Nuclear extracts and cytosolic fractions were prepared and analyzed for RUNX3, phospho-Ser249-AML1 (pAML1), AML1, AP-2α, HDAC1 and GAPDH levels. Values are the mean ± SD of four independent experiments. *** *p* < 0.001 vs. control (vehicle) by Student’s *t* test. (**B**) SH-SY5Y cells transfected with either control siRNA or RUNX3 siRNA duplexes were incubated for 24 h with either vehicle, 0.6 mM VPA (VPA 0.6), or 1 mM VPA (VPA 1.0). Cell lysates were analyzed for RUNX3 and TrkC expression. Values are the mean ± SEM of four independent experiments. *** *p* < 0.001 vs. control (control siRNA + vehicle). ^#^
*p* < 0.05, ^###^
*p* < 0.001 vs the corresponding value in control siRNA-treated cells. (**C**) SH-SY5Y cells were incubated for 72 h in growth medium with either vehicle, 1 µM 3-deazaneplanocin A (DZNep), or 1 µM tazemetostat (TZM). Cell lysates were analyzed for TrkC and RUNX3 levels. Values are the mean ± SD of four independent experiments. *** *p* < 0.001 vs. the corresponding control value (vehicle) by ANOVA followed by Tukey’s test.

**Figure 5 ijms-22-07790-f005:**
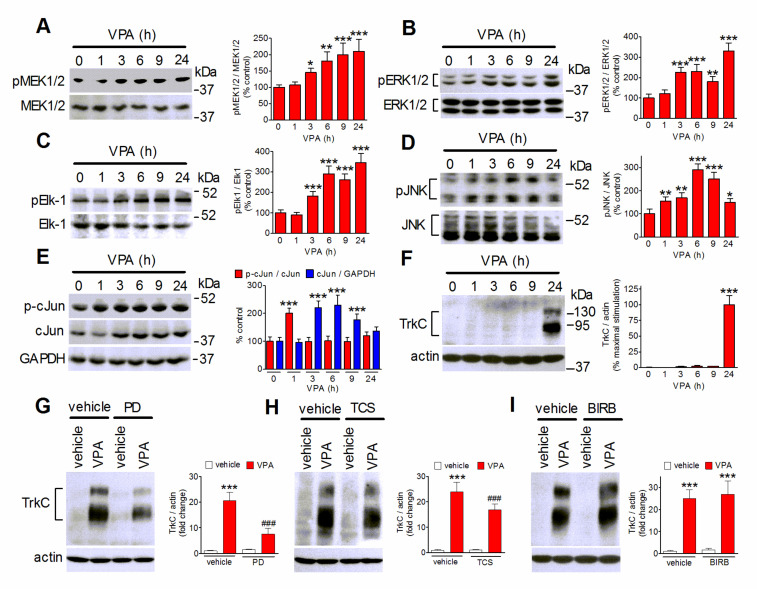
Participation of ERK1/2 and JNK in VPA-induced TrkC upregulation. (**A**–**F**) SH-SY5Y cells were incubated in serum-free medium with 1 mM VPA for the indicated periods of time. Zero time samples were treated with vehicle and used as control. Cell lysates were analyzed for phospho-MEK1/2 (pMEK1/2), MEK1/2 (**A**), phospho-ERK1/2, ERK1/2 (**B**), phospho-Ser383-Elk-1 (pElk-1), Elk-1 (**C**), phospho-JNK (pJNK), JNK (**D**), phospho-Ser73-cJun (p-cJun), cJun, GAPDH (**E**), TrkC and actin (**F**). Values are the mean ± SD of four independent experiments. * *p* < 0.05, ** *p* < 0.01, *** *p* < 0.001 vs. control. (**G**–**I**) SH-SY5Y cells were preincubated for 1 h with either vehicle, 25 µM PD 98,059 (PD) (**G**), 10 µM TCS JNK6o (TCS) (**H**), or 1 µM BIRB 0796 (BIRB) (**I**) and then exposed to either vehicle or 1 mM VPA for 24 h. Cell lysates were analyzed for TrkC expression. Values are the mean ± SD of four experiments. *** *p* < 0.001 vs. control (vehicle + vehicle); ^###^
*p* < 0.001 vs. vehicle + VPA by ANOVA followed by Tukey’s test.

**Figure 6 ijms-22-07790-f006:**
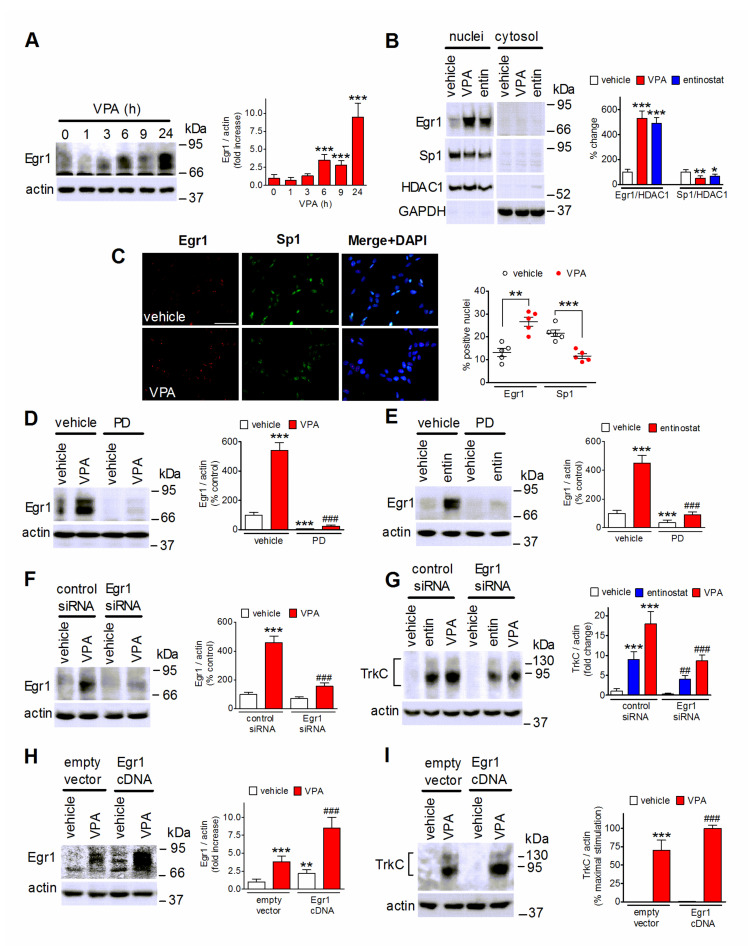
Induction of Egr1 mediates TrkC upregulation by VPA. (**A**) SH-SY5Y cells were incubated in the presence of 1 mM VPA for the indicated periods of time. Zero time samples (control) were treated with vehicle. Cell lysates were analyzed for Egr1 and actin. Values are the mean ± SD of four experiments. *** *p* < 0.001 vs. control. (**B**) SH-SY5Y cells were treated for 24 h with either vehicle, 1 mM VPA, or 1 µM entinostat (entin). Nuclear extracts and cytosolic fractions were analyzed for Egr1, Sp1, HDAC1 and GAPDH levels. Densitometric ratios of Egr1/HDAC1 and Sp1/HDAC1 in nuclear extracts are reported as percent change with respect to control (vehicle) and are the mean ± SD of four independent experiments. * *p* < 0.05, ** *p* < 0.01, *** *p* < 0.001 vs. control by ANOVA followed by Tukey’s test. (**C**) SH-SY5Y cells were incubated for 24 h with either vehicle or 1 mM VPA and then analyzed for Egr1 and Sp1 expression by immunofluorescence. Nuclei were stained in blue with DAPI. Scale bar, 50 µm. Values are the mean ± SD of five independent experiments. ** *p* < 0.01, *** *p* < 0.001 vs. the corresponding control (vehicle) by Student’s *t* test. (**D**,**E**) SH-SY5Y cells were preincubated for 1 h with either vehicle or 25 µM PD 98,059 and then exposed for 24 h to either vehicle, 1 mM VPA (**D**), or 1 µM entinostat (**E**). Cell lysates were analyzed for Egr1 expression. Values are the mean ± SD of four independent experiments. *** *p* < 0.001 vs. control (vehicle + vehicle); ^###^
*p* < 0.001 vs. vehicle + VPA. (**F**) SH-SY5Y cells transfected with either control siRNA or Egr1 siRNA were treated for 24 h with either vehicle or 1 mM VPA. Cell lysates were analyzed for Egr1 levels. Values are the mean ± SD of four independent experiments. *** *p* < 0.001 vs. control siRNA + vehicle; ^###^
*p* < 0.001 vs. control siRNA + VPA. (**G**) SH-SY5Y cells were transfected as in (**F**) and then exposed for 24 h to either vehicle, 1 µM entinostat or 1 mM VPA. Cell lysates were analyzed for TrkC expression. Values are the mean ± SD of four independent experiments. ^***^
*p* < 0.001 vs. control siRNA+ vehicle; ^##^
*p* < 0.01, ^###^
*p* < 0.001 vs. the corresponding sample in control siRNA-treated cells. (**H**,**I**) SH-SY5Y cells transfected with either empty vector or Egr1 cDNA were treated for 24 h with either vehicle or 1 mM VPA. Cell lysates were analyzed for Egr1 (**H**) and TrkC (**I**) expression. Values are the mean ± SD of four independent experiments. ** *p* < 0.01, *** *p* < 0.001 vs. empty vector + vehicle; ^###^
*p* < 0.001 vs. empty vector + VPA by ANOVA followed by Tukey’s test.

**Figure 7 ijms-22-07790-f007:**
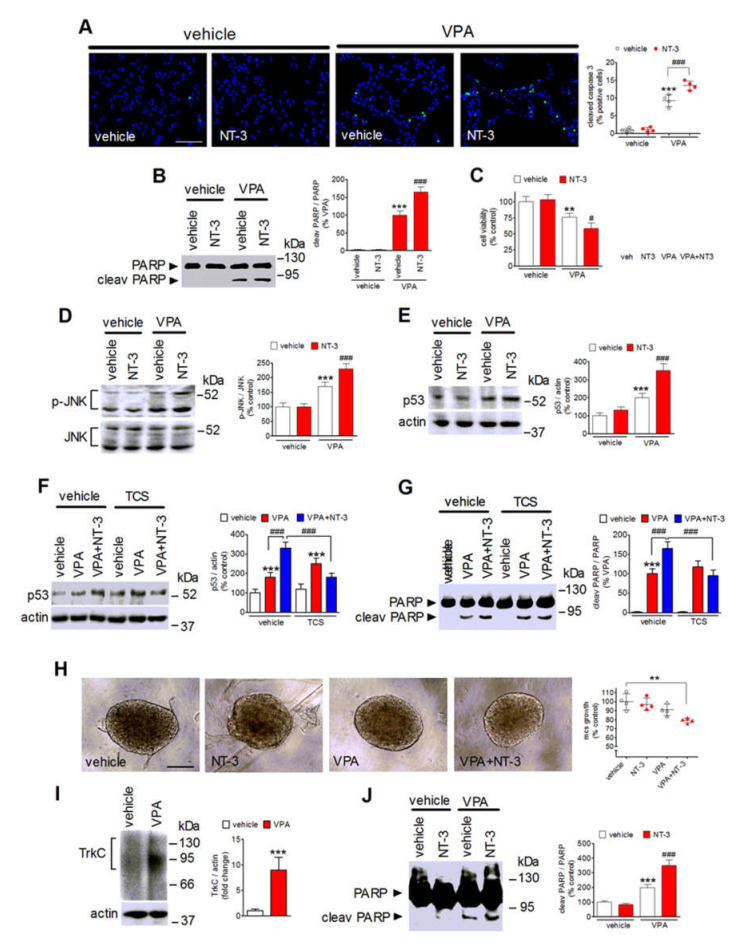
NT-3 potentiates VPA-induced apoptotic cell death. (**A**) SH-SY5Y cells were incubated with either vehicle or 1 mM VPA for 24 h and then exposed for 24 h in serum-free medium to either vehicle or 0.3 nM NT-3. Cells were analyzed for cleaved caspase 3 (green color) by immunofluorescence microscopy. Nuclei were stained in blue with DAPI. Positive cells are expressed as percent of total cells. Scale bar, 50 µm. (**B**) Cells were treated as in (**A**) and cell lysates were analyzed for PARP cleavage. (**C**) Cells were treated as in (**A**) and then analyzed for viability by using Real Time-GLO MT assay. (**D**,**E**) Cells were treated as in (**A**) and cell lysates were analyzed for phospho-JNK and p53 expression. Values are the mean ± SD of four independent experiments. ** *p* < 0.01, *** *p* < 0.001 vs. control (vehicle + vehicle); ^#^
*p* < 0.05, ^###^
*p* < 0.001 vs. VPA + vehicle. (**F**,**G**) Cells pretreated for 24 h with either vehicle or 1 mM VPA were incubated for 1 h with either vehicle or 10 µM TCS JNK6o (TCS) and then exposed to either vehicle or 0.3 nM NT3 for an additional 24 h. Cell lysates were analyzed for p53 levels and PARP cleavage. Values are the mean ± SD of four independent experiments. *** *p* < 0.001 vs. control (vehicle + vehicle); ^###^
*p* < 0.001. (**H**) Light microscopy images of SH-SY5Y multicellular spheroids (mcs) pretreated for 24 h with either vehicle or 1 mM VPA and then exposed for an additional 48 h to either vehicle or 1 nM NT-3. The mcs size of each experimental group is reported as percent of control (vehicle + vehicle). Scale bar, 200 µm. Values are the mean ± SD of four individual experiments. ** *p* < 0.01. (**I**) Spheroids of SH-SY5Y cells were treated for 48 h with either vehicle or 1 mM VPA and then analyzed for TrkC expression by Western blot. Values are the mean ± SD of four individual experiments. *** *p* < 0.001 vs. control (vehicle) by Student’s *t* test. (**J**) Spheroids of SH-SY5Y cells were treated as in (**H**) and analyzed for PARP cleavage by Western blot. Values are the mean ± SD of four individual experiments. *** *p* < 0.001 vs. control (vehicle + vehicle); ^###^
*p* < 0.001 vs. VPA + vehicle by ANOVA followed by Tukey’s test.

**Figure 8 ijms-22-07790-f008:**
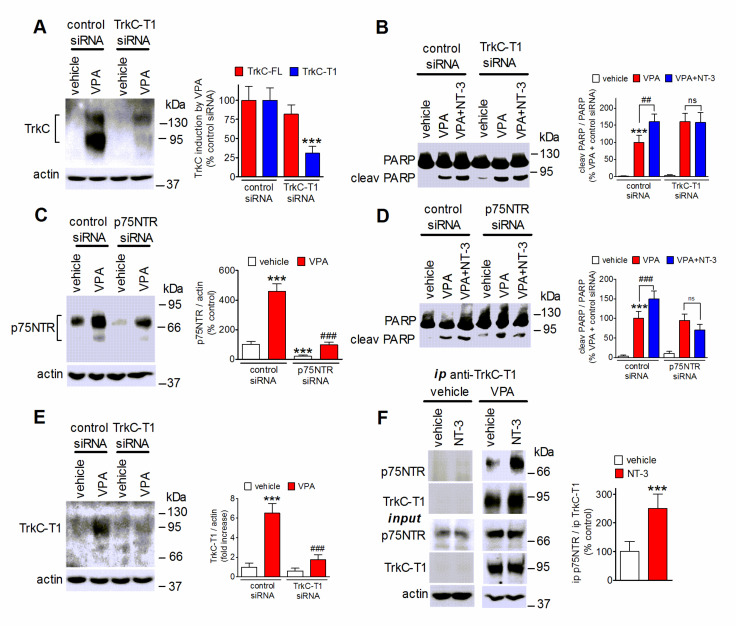
TrkC-T1 and p75NTR mediate NT-3 potentiation of VPA-induced apoptosis. (**A**,**C**) SH-SY5Y cells were transfected with either control siRNA, TrkC-T1 siRNA, or p75NTR siRNA and then treated for 24 h with either vehicle or 1 mM VPA. Cell lysates were analyzed for TrkC-FL and TrkC-T1 (**A**), and p75NTR (**C**). Values are the mean ± SD of four independent experiments. *** *p* < 0.001 vs. control (control siRNA + vehicle); ^###^
*p* < 0.001 vs VPA in control siRNA-treated cells. (**B**,**D**) Cells transfected as in (**A**,**C**) were preincubated for 24 h with either vehicle or 1 mM VPA and then exposed in serum-free medium to either vehicle or 0.3 nM NT-3 for 24 h. Cell lysates were analyzed for PARP cleavage. Values are the mean ± SD of four independent experiments. *** *p* < 0.001 vs. control (control siRNA + vehicle); ^##^
*p* < 0.01, ^###^
*p* < 0.001; ns, *p* > 0.05. (**E**) Cells were transfected and treated as in (**A**) and cell lysates were analyzed for TrkC-T1 expression by using a TrkC-T1 specific antibody. Values are the mean ± SD of four independent experiments. *** *p* < 0.001 vs. control (control siRNA + vehicle); ^###^
*p* < 0.001 vs. VPA in control siRNA-treated cells by ANOVA followed by Tukey’s test. (**F**) SH-SY5Y cells pretreated for 24 h with either vehicle or 1 mM VPA were then exposed for 1 h to NT-3 (1.0 nM) in serum-free medium. Cell extracts were then subjected to immunoprecipitation (ip) with anti-TrkC-T1 antibody. Cell extracts (input) and immunoprecipitates were analyzed for p75NTR and TrkC-T1 levels. Values refer to the densitometric ratios of p75NTR over TrkC-T1 in immunoprecipitates of VPA-treated cells and are the mean ± SD of four independent experiments. *** *p* < 0.001 vs. control (VPA + vehicle) by Student’s *t* test.

**Figure 9 ijms-22-07790-f009:**
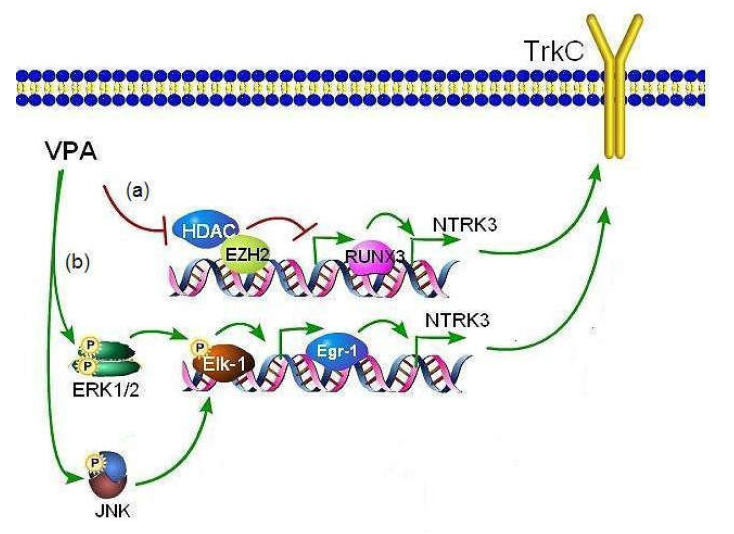
Putative molecular mechanisms mediating the induction of TrkC expression by VPA in human neuroblastoma cells. The diagram illustrates that VPA enhances the expression of the NTRK3 gene encoding TrkC through the following mechanisms: (**a**) epigenetic changes involving RUNX3 derepression as the consequence of HDAC inhibition and EZH2 depletion [32,42]; (**b**) activation of ERK1/2 and JNK signaling pathways. ERK1/2 activation exerts a major stimulatory input on NTRK3 activation by inducing Egr1 expression likely via Elk-1 phosphorylation. The stimulation of JNK appears to provide a minor contribution possibly by participating in Elk-1 phosphorylation.

## Data Availability

Data are contained in the article and Appendix A.

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
