# Peer review of "The Neurotrophin Receptor TrkC as a Novel Molecular Target of the Antineuroblastoma Action of Valproic Acid"

_ijms, 2021, doi:10.3390/ijms22157790_

Round 1

Reviewer 1 Report

Simona et al. suggested novel mechanism of antineuroblastoma action of VPA, a broad spectrum HDAC inhibitor. The authors disclosed VPA-treatment resulted in activation of ERK1/2 and JNK signal to induce Egr1 and thereby transcription of RUNX3. TrkC was expressed by RUNX3 and recruited p75NTR in response to NT-3, which finally activated caspase 3 to induce apoptosis of neuroblastoma cells. Although these results show massive amount of signal cascades, the reviewer suggests several concerns to be addressed for further consideration.

Major points

  1. Fig 6 is missing
  2. Provide working hypothesis to summarize all signal cascades
  3. The authors concluded many part of their finding on the basis of inhibitor study or siRNA study. However, as the author might agree with this points, chemical inhibitor could not represent specific signal cascade. Those drugs may affect diverse signal simultaneously. The reviewer suggests that the author would include transfection study: which include RUNX3 and Egr1.
  4. Although the authors achieved successful, almost complete when compared to vehicle treatment in Fig4B, knockdown of RUNX3 with its specific siRNA, TrkC level was not fully reduced. This unexpected data may implicate that there is a bona fide mechanism for TrkC transcription beyond RUNX3.
  5. According to BET inhibitor study, histone acetylation and BRDx recruitment is mandatory for TrkC transcription. The reviewer suggests making signal cascades as concise as possible. When consider “VPA” is HDAC inhibitor, activation in early time points, such as 1 hr ~ 6 hr, may implicate off target effects rather than histone modification. Are they modified by acetylation-dynamic mechanisms?

Minor points

  1. In IMR32 cells, full length of TrkC are more abundant than truncated form. Can the authors explain this discrepancy?
  2. The reviewer could not agree that the changes of p-cJun/cJun in Fig5E
  3. Why the basal level of protein expressions is so variable? For example, RUNX3 of Fig4A and 4B is different and p75NTR of Fig8C (control siRNA) and 8F (vehicle) are totally different.
  4. Several WB images are not acceptable quality for paper. Improve them. Beside, several images are fully saturated (i.e. Fig 8B). Do not exaggerate contrast to maximize the effect.
  5. According to description in method sections, the authors mainly applied U test for statistical analysis. However, Kruskal-Wallis and Boferroni correction must be utilized for non-parametric comparison between more than two group. Use appropriate statistical method. The reviewer’s comment includes Fig 2D, 2E, all of fig 3, 4B, 4C, all of fig 5, all of fig 7, 8A, 8B, 8C, 8D, 8E. Instead of nonparametric comparison, the author may applicate ANOVA for parametric test when the data satisfy normal distribution.

Reviewer 2 Report

The article by Dedoni S., et al describes that VPA upregulates TrkC by activating epigenetic mechanisms and signaling pathways and sensitizes neuroblastoma cells to NT-3 induced apoptosis. The authors also show that cell monolayers and spheroids the exposure to NT-3 enhanced the apoptotic cascade triggered by VPA. The findings are interesting and could have important clinical applications. However, the exact message of how epigenetic mechanism controlling the expression of membrane receptor protein is not clear. Authors would have showed in vivo phenotype and in vitro overexpression data alongside siRNA which are the major drawback of this study.  

My main concerns are:

  1. Please comment how the epigenetic changes are contributing to the surface expression of TrkC-FL/T1 expression and observed phenotype.
  2. How the concentration of the VPA and other drugs in the study are decided, the reason is that just one concentration of VPA (1mM) and maximum of 24 h was done in most of the experiments. What is the rational for using this concentration and not looking at the expression of TrkC-FL/T1 beyond 24 h time point?
  3. Fig 6 is missing in the manuscript. Please include in rebuttal.
  4. In Fig1. Why would there be no basal expression of TrkC-FL (other than IMR 32). Is this normal or please include the relevant blots.
  5. In section 2.4 authors mentioned “undifferentiated SH-SY5Y cells treated with VPA” why is undifferentiated so specifically mentioned?
  6. Authors would have performed overexpression studies alongside siRNA. This would have made the observed data much stronger.
  7. In Fig 7H please include clear spheroid images. Quantification and represented image are not matching.
  8. A summary figure can aid in better understanding of the concept. Please make a summary figure.

Minor comment

  1. Authors would have represented dot plots representation of the data presented in the manuscript.

Round 2

Reviewer 1 Report

The author gave an appropriate answer to the problemthe reviwer presented, and the manuscript was also appropriately revised.